# University Technology Transfer from a Knowledge-Flow Approach—Systematic Literature Review

**José Bestier Padilla Bejarano** [1,2,*], **Jhon Wilder Zartha Sossa** [3], **Carlos Ocampo-López** [4] **and Margarita Ramírez-Carmona** [4]

1. Program in Electronic Instrumentation Technology, Faculty of Basic Sciences and Technologies, Universidad del Quindío, Armenia 030004, Colombia
2. Doctoral Program in Management of Technology and Innovation, Universidad Pontificia Bolivariana, Medellin 050031, Colombia
3. Department of Engineering, Faculty of Agro-Industrial Engineering, Universidad Pontificia Bolivariana, Medellín 050031, Colombia
4. Centro de Estudios y de Investigación en Biotecnología (CIBIOT), Faculty of Chemical Engineering, Universidad Pontificia Bolivariana, Medellín 050031, Colombia
* Correspondence: jbpadilla@uniquindio.edu.co; Tel.: +57-311-7674009

**Abstract:** This study aims to review and synthesize the rapidly evolving literature on technology transfer from universities, and the concepts and models included in it, from a knowledge-flow approach to find the factors contributing to its performance. This article provides a perspective on recent work, focusing on empirical studies on technology transfer in universities conducted in the last 32 years from a knowledge-based vision. The study was carried out from a systematic literature review in the Scopus, Web of Science, ScienceDirect, and SpringerLink databases on 135 articles selected and evaluated by peers from critical surveillance factors such as technology transfer, knowledge flow, and university–industry relationship, among others. It was possible to identify 75 factors that, from a knowledge-based vision and specifically from a knowledge-flow approach, permit and contribute to the performance of technology transfer generated from research processes in universities. We classified studies into four categories according to their approaches, each with their dimensions and factors: management of knowledge, resources and capabilities, management of technological transfer, and the university–industry relationship. This classification permitted not only identification but also the systematization of the different factors and related authors that, from a knowledge-flow approach, contribute to the performance of technological transfers in universities, reflecting their efficiency and effectiveness. In this respect, absorption capacity and open innovation are topics which are worthy of exploration.

**Keywords:** knowledge flow; research results; technological transfer; university; university–industry relationship

## 1. Introduction

Industry evolves continuously, reflecting changes aimed at a society based on knowledge and services. This has provoked an extensive discussion in the literature on the university's role in, and contribution to, the economy [1]. Thus, seeking an elevated dynamism in the environment, the construction of collaborative settings promoting innovation is a key element for developing regional competitiveness. This is how the relationship between the industry, university, and government is strengthened, changing the approach to the processes of technology transfer from university to industry [2]. This triple relationship plays a strategic role by generating technological knowledge which is transferable to industry. It can be transformed into economic and social value for users, clients, the institution, and society [3].

The knowledge and technology transfer processes that take place in universities and that end up having an impact on the productive sectors have led to an increase in academic research on the subject of technology transfer [4], given that it is widely recognized that universities contribute to the social, economic, and cultural development of the regions in which they operate [5]. Although universities are dedicated to teaching and research activities, the trend is that they become involved in a "third mission", which is understood as a contribution to society through knowledge and technology transfer activities [4,5] and which contributes significantly to innovation processes in the productive sector, where technological innovation is achieved through the active flow of knowledge among knowledge actors [6–8].

In fact, technology transfer from universities to industry is considered a competitive strategy, given that business development can be boosted by academic research providing new scientific discoveries and advanced technologies that accelerate innovation [7–9]. While knowledge transfer promotes comprehension of what caused a change, technology transfer points to the means for change. Effective knowledge transfer is associated with higher productivity, survivability, and competitive advantage [3]. Thereby, emerging technological advances and developments which were started years ago in universities can transform business models, including processes and mechanisms of innovation and governance, structures and roles, systems of relationships, and limits of companies [2].

The industry–university–government link facilitates innovation development directly proportional to the increased stakeholder rating, with each axis element being interchangeable [10]. Thus, some authors focus on the analysis of technological transfer from universities to the actors and elements that appear in the complex university–industry relationship [11–22]. For their part, [19,23] relate the channels and mechanisms present in technology transfer, while [12,21,23–29] describe the barriers occurring in university–business relationships and how they affect technology transfer. This last approach supports that several identified factors affect performance when transferring technology from inside universities to outside them. However, much of the existing literature makes no explicit reference to studies involving the university–industry relationship from a knowledge-flow approach, an approach which is characterized by two fundamental aspects: the content of the knowledge transferred (reflected in the technologies to be transferred) and the direction of the knowledge flow (the actor responsible for initiating the university–industry interaction) [30]. With these two characteristics, it is possible to determine how the dynamic flow is produced during the university–industry interaction and how this flow stimulates innovative activities and, hence, technology transfer.

The work then proposed identifying and analyzing the most relevant factors that, from a knowledge-flow approach, could contribute to the performance of the transfer of research results from universities to the production sector or society. This way, it is expected to contribute to current studies of technological transfer from Higher Education Institutions, promoting a reference framework concerning local, national, and international experiences.

The article is divided into several sections. The first presents some concepts concerning technological transfer from universities and recent studies about knowledge flow and its contribution to the performance of the said transfer; thereafter, the methodology applied is exposed in three phases, starting from the selection of the database used, keywords, criteria, search equations, reading and discussion of the 135 selected documents. Then, the results from reading the documents, from the identification and selection of the variables with their dimensions, and the coding of the study factors are presented, as well as an analysis of the application of the VOSviewer version 1.6.11 software with an emphasis on authors and keywords. In the end, a discussion of the analysis of the factors is presented, with their evolution and their impact on the performance of technological transfer from Higher Education Institutions to the production sector or society.

This study permitted the different factors and related authors that, from a knowledge-flow approach, have contributed to the performance of technological transfer in universities, to be identified and systematized, reflecting their efficiency and effectiveness. Likewise,

the study identified the authors who have led the generation of the new methods, models, and strategies that permit the evolution of the technology transfer process to continue. It is highlighted that concepts such as absorption capabilities and open innovation gain more importance each day in the university–industry relationship and, thereby, in the knowledge flow taking place between these players.

## 2. Theoretical Framework

The concept of technology transfer has been used by various disciplines to describe and analyze various processes and factors related to technological developments and their dissemination. The study and analysis of this concept in universities have been addressed from different theoretical approaches, allowing models and strategies that strengthen the university–industry relationship to be described and generated. The following describes elements that enhance the study conducted.

### 2.1. Knowledge Transfer (KT)

Culter [31] defines knowledge transfer between two or more players (individuals or organizations) as the process through which a social actor acquires knowledge from another actor. Liao and Hu [32] describe it as the process through which receptors accumulate and renew the production capacity from knowledge received, while Barraza [33] defines it as the process through which knowledge is transmitted by one actor and absorbed by another to improve their capabilities.

### 2.2. Technology Transfer (TT)

During the 1960s and 1970s, this term emerged as an aid to manage and improve the economic development of countries where TT and its contractual dimension are at the forefront among developed and developing countries [34].

Bozeman defines technology and TT in diverse manners [12], where each definition depends on the discipline and the research purpose [35]. Lundquist [36] defines TT as passing a technique or knowledge developed in one organization to another where it is adopted and used. One of the definitions the authors of this study consider as the most adequate for the concept of TT is formulated by the Association of University Technology Managers AUTM [37] "Technology transfer is the process of designation of the formal transfer to the industry of discoveries resulting from the university or private research, for commercialization purposes under the form of new products and/or services" [38] (p. 2).

Contreras [39] argues that the concept of "technology transfer" is composed of two elements that together try to increase the productivity and competitiveness levels of organizations; the first term, "transfer," is conceived as the combination of giving and receiving information (giving or receiving knowledge) from one unit of value generation to another, to enhance the production activity by providing mechanisms that permit adaptation to the external environment; "technology" corresponds to the methodical application of technical knowledge to generate new products and/or services or their respective improvement.

Within the context of TT from the public sector and universities to the private sector, the World Intellectual Property Organization (WIPO) indicates that the term "technology transfer" is synonymous with "commercialization of technology" through which the results of primary scientific research carried out by universities and public research organisms on commercial and practical products from private companies, which are destined for market, are applied [40].

For López et al. [41] TT is understood as "the process through which the private sector gets access to technological advances developed by scientists, through the transfer of said developments to production companies for their transformation into useful goods, processes, and services commercially usable" (p. 72). Within this context, TT is considered a link between the university and companies that dynamizes the generation of scientific, technical, and economic development. López et al. [41] indicate that the transfer entails a

convention and an agreement, and presupposes a payment; hence, the commercialization of knowledge is an element inherent in this process.

It should be indicated that distinctions exist in the literature between knowledge transfer and technology transfer [42]. Bozeman [12] determines knowledge transfer as "the scientific knowledge used by scientists to advance in science" and technology transfer as "the scientific knowledge used by scientists and others in new applications", highlighting that the latter has received the most attention in the literature on TT (p. 642). For Amesse and Cohendet [34], TT is a specific knowledge transfer process that depends on how companies manage knowledge.

### 2.3. University–Industry Relationship

These relations consist of the presence of a knowledge creation actor (university) together with another actor responsible for applying that knowledge to technology and innovation (industry) [43]. To generate an adequate knowledge flow between universities and industry, it is necessary to establish continuous relationships, such as long-term projects [44]. Government agencies promote these relationships through regulatory laws, group organization, education, and research incentives [45].

The relationship between the university and industry is increasingly more significant, generating the need to create and increase, in some cases, bodies or departments in charge of managing knowledge that allow advisory services, consultancies in technical areas, technology licensing agreements, technical assistance, training programs and specific technological development contracts to be provided [35]. These entities generate specific mechanisms to negotiate, administer projects, and structure contracts for managing knowledge and technology [11,23,46–49].

### 2.4. Knowledge Flow

Knowledge flow is defined as a process of knowledge transmission between people or knowledge processing mechanisms. It is characterized by three essential attributes: direction (sender and receiver), bearer (transmission means or channel), and content (that which will be transmitted or shared) [50]. Zhang and Li [51] defined knowledge flow as a process of production, transfer, and application of knowledge among various participants. From the perspective of technological transfer, Dalmarco et al. [30] define the term knowledge flow using two aspects: direction and content of knowledge. Direction relates to the actor responsible for proposing the interaction through stimuli, whether the university, industry, or government. At the same time, knowledge content is associated with the technology production expected to result from the association. This could be a new technology based on fundamental research or a combination of existing technologies, with it being the actor responsible for the interaction who defines the knowledge content.

To promote knowledge flow and TT, it is essential to have participation from governments, academic institutions, scientific research centers, and the production sector and society to explore and improve an effective mechanism of the technology to be transferred [44].

## 3. Materials and Methods

The methodology was conducted in three phases; based on the systematic literature review, Table 1 describes the phases, and the information on each one is expanded.

**Table 1.** Systematic literature review and methodology established.

| Phases | Description | Approach of the Review |
|---|---|---|
| Phase I | Define the purpose and objective of the review | Review prior studies on technology transfer in universities, and search for factors related to knowledge flow. |
| | Search strategy | Use of critical surveillance factors (CSF) to establish the search equation—selection of specific databases. |
| Phase II | Criteria used to select and include review sources | The following are the selection criteria for the review:<br>• Articles in peer-reviewed journals with theoretical studies<br>• Studies on technological transfer in universities<br>• Articles containing themes related to the categories or conceptual axes chosen from the axial coding criteria |
| | Conditions used to omit publications during the review process | The exclusion criteria for the review were:<br>• Duplicate articles<br>• Articles not related to topics regarding the categories or conceptual axes chosen from the axial coding criteria. |
| Phase III | Information grouping and analysis | Analysis and interpretation of findings |

Phase I. Upon defining the purpose of the study was to review and synthesize the literature on technology transfer propitiated by universities and its concepts and models from a knowledge-flow approach to find the factors that contribute to its performance, a systematic literature review was carried out, initially on 250 articles, in Scopus, Web of Science, Science Direct and SpringerLink during the period from 1990 to 2022. The critical surveillance factors were established using the search keywords: technological transfer, knowledge flow, higher education institutions, and university. The review of articles focused on the lines of discussion emphasized in each article (knowledge management, technological transfer management, resources, capabilities, and the university–industry relationship). The following search equations were implemented to guarantee the proximity of the key terms:

1. TITLE-ABS-KEY ("technology transfer" AND "knowledge flow" AND ("university" OR "higher education institutions"))
2. TITLE-ABS-KEY ("technology transfer" AND "knowledge-based vision" AND ("university" OR "higher education institutions"))

Phase II. The abstracts, keywords, and conclusions were analyzed to select and/or omit the articles found and to guarantee that they were related to the search terms and the study approach. As a result of this process, 135 articles were found to have a direct relation to technology transfer in universities and to have elements that related to factors from a knowledge-flow approach.

After reading the 135 articles, an analysis was performed to categorize the axes or conceptual groups on which the most significant contributions of the articles were found; these axes were chosen according to the axial coding criteria proposed by Hernández et al. [52]. The factors selected (52 in all) were grouped into dimensions belonging to a category or specific variable related to the content and direction of the knowledge flow [30,44,51].

Phase III. With the 52 factors identified in the review of the 135 publications, a conceptual framework was made concerning the variables, dimensions, and factors that permit the theoretical models, concepts, arguments, and ideas developed concerning the technological transfer to be analyzed. In the same way, an analysis was carried out based on the information provided by VOSviewer in relation to the map of the occurrence of the keywords and the network of co-authors of the selected papers.

Figure 1 illustrates the systematic reviews in a PRISMA 2020 flow diagram [53] in which the searches in databases and number of records are included.

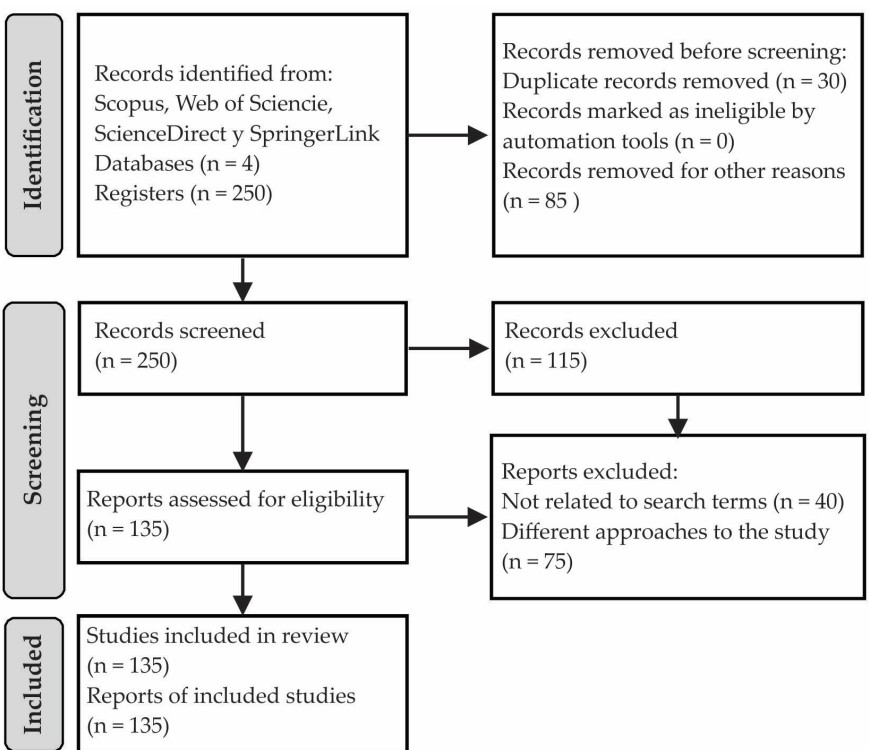

**Figure 1.** Identification of studies via databases and registers. Source: elaborated by the authors based on [53].

## 4. Results

The following presents the results obtained in the review of the 135 articles, where it was possible to identify the variables, their dimensions, and 75 relevant factors that influence the performance of technological transfer taking place in universities from a knowledge-flow approach [30,44].

Figure 2 illustrates the conceptual axes or groupers that allowed the factors to be classified into their respective dimensions.

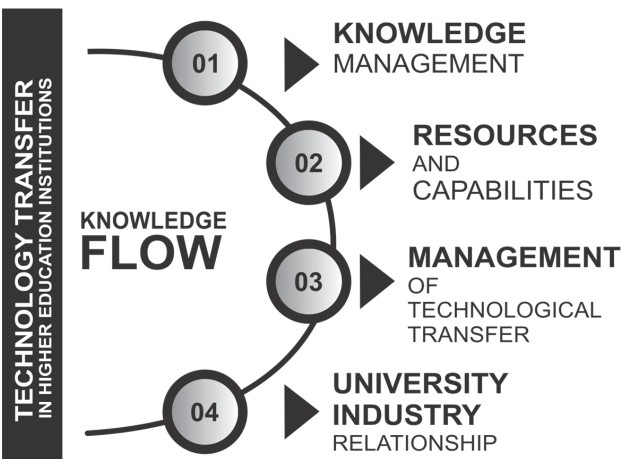

**Figure 2.** Conceptual axes or groupers. Source: Elaborated by the authors.

For Avendaño and Flores [54], knowledge management emerges as a management approach or emerging discipline which, in a structured and systematic manner, seeks to

avail of the knowledge generated in the organization to achieve organizational objectives and optimize the decision-making process. Table 2 illustrates the factors and dimensions that—from knowledge management—could contribute to technological transfer. Likewise, it mentions some authors that, from the theory, have contributed to the creation [28,55,56], storage, and transfer of knowledge [56–59], as well as to its application and use [57,60–62].

**Table 2.** Dimensions and factors related to knowledge management.

| Dimensions | Factors | Authors |
| --- | --- | --- |
| Creation (organizational learning) | Generation of knowledge | [55,56,58] |
| | Infrastructure and tools for information analysis | [28,56] |
| Storage and transfer (organizational knowledge) | Dissemination of knowledge | [56,58] |
| | Social approach | [54,57,59,63–66] |
| | Economic approach | [21,24,29,54,57–59,65] |
| Application and use (organization of learning) | Appropriation of knowledge | [57,61,62,67–70] |
| | Commercialization of knowledge | [56,60] |

From a modern administrative thought-based approach, it is indicated that resource-based theory is considered the best form of organization, given that it permits its resources and capabilities to be managed more rationally [71,72]. This theory establishes that having resources within a company is valuable, and that they must be difficult to imitate, unique, and non-substitutable. Likewise, it suggests that organizations must concentrate their efforts within the company to find sources of competitive advantage by using their resources. This theory emphasizes the organization's internal resources and the availability of productive services from their own resources, particularly those from management with experience within the company [73]. As stated by Amit & Schoemaker [74], Grant [75], and Penrose [76], resources are the set of physical and human inputs an organization has, and—through these—its activities and tasks are carried out (capital teams, employee skills, patents, brands).

Table 3 mentions the factors and their dimensions that, from resource-based theory, contribute to the performance of technology transfer in universities. It is highlighted that within resource-based theory, the key factors include tangible and intangible resources [12,21,61,74,76–78] and capabilities (basic capabilities in R&D, organizational and dynamic capabilities) [62,64,70–72,79,80].

**Table 3.** Dimensions and factors from the resource-based theory.

| Dimensions | Factors | Authors |
| --- | --- | --- |
| Tangible Resources | Investment in R&D | [7,21,61,74–76,78,81,82] |
| | Adequate R&D infrastructure | [7,21,61,73–76,78,82,83] |
| Intangible Resources | Trained human capital | [21,61,82–86] |
| | Intellectual property | [12,26,65,87,88] |
| | Normative-regulatory framework | [15,81,83,89] |
| | Internal policies | [56,90] |
| | Intellectual property rights | [65,77,78,80] |
| Basic capabilities in R&D | R&D integration mechanisms | [64,85,91] |
| | Internal capabilities of the university | [21,56,58,78,80,92,93] |
| Organizational capabilities | Generation of R&D competitive advantages | [54,61] |
| Dynamic capabilities | Knowledge absorption capacity | [34,62,69–72,79] |

For Grant [75], capacity is the sufficiency or ability of a team of resources to perform a task or activity and constitutes the primary source of competitive advantage, just as resources are the sources of a company's capabilities. The theory of resources and capabilities is fundamentally valid and continues to evolve to the extent that we speak of dynamic capabilities [62,70], which implies adapting to change in order to integrate, construct, and reconfigure internal and external resources. Teece [94] illustrates how innovation in organizations has been enriched by identifying dynamic capabilities, the role of knowledge, and knowledge flow as sources of these capabilities [69] and patent citations as a statistical tool to capture this knowledge flow [68,95,96].

In relation to technological transfer management, many authors have contributed to the performance of this transfer from different dimensions; the approach of this article kept in mind those dimensions related to the R&D process [80] and the conditions of the environment [16,23,56,57,77,97]. From the different models proposed for technological transfer in universities based on the university–industry relationship and/or from knowledge-based vision [23,47,48,54,64,98], some authors described interesting factors that were considered in relation to the management of technological transfer and which are displayed in Table 4.

**Table 4.** Dimensions and factors related to technology transfer management.

| Dimensions | Factors | Authors |
|---|---|---|
| Research, Development, and Transfer process | Motivation | [78,80,86] |
| | Planning of the transfer process | [12,99,100] |
| | Formalization of the R&D + i process | [78] |
| | Transfer modalities | [99,101,102] |
| | Culture of innovation | [66,77,103] |
| | Interaction among the players in the innovation system | [21,23,77,102,104] |
| | Transfer models | [14,16,19,33,41,97,105–113] |
| | Development state of R&D + i, susceptibility to being transferred | [12] |
| | Measurement | [114] |
| | Creation of new technological-based enterprises | [49,65,115] |
| | Technological surveillance that avoids duplicating efforts | [56] |
| | Execution of research, development, and transfer processes | [99,100,104,116,117] |
| Conditions of the environment (organizations, state, and society) | State incentives to facilitate knowledge flow | [58,118] |
| | Role of society in the university–industry relationship | [64,77] |
| | Conditions of enterprises regarding development of R&D + i stemming from academia | [26,41,77,78,87] |
| | Conditions of the state concerning development of R&D + i stemming from academia | [57,77,119] |
| | Vocation of academic players to find solutions to concrete productive/social problems | [23] |
| | Non-traditional incentives for performance for university personnel and R&D centers | [25,56] |
| | Leadership | [56,97] |
| | Conditions of society regarding development of R&D + i stemming from academia | [77,78,87,88,120] |

Other theories that have strengthened technological transfer from universities are analyzed according to how the university–industry relationship takes place, from the relationship described in the triangle by Sabato [121] to the various models or modes of the triple helix [15,16,106] and its evolution to the quadruple and quintuple helix [122,123]. Sábato's triangle explains how each vertex: government, industry, and science (scientific-technological infrastructure) interacts with each other or with society in a one-way information flow, while the Triple helix describes a dynamic interaction among the same three vertices in which government establishes policy, and industry and science interact constantly. Besides universities and enterprises, the government is also an essential part of the tripod of University–Industry (U-I) relations, principally due to laws, policies, and funds [16,97,124–129]; the triple helix is considered an expansion of the role of knowledge in society and of the university in the economy [130,131]. These theories describe the players involved in university–industry relations and knowledge transfer channels. According to the country's environment, universities or enterprises establish different forms of knowledge transfer depending on the channels used.

Bercovitz and Feldman [105] add further to our understanding of the relationships between university and industry and their role in knowledge-based innovation systems. These authors highlight that in addition to the legal, economic, and policy environments that make up an innovation system, internal influences may exist within a university that determine the rates and directions of knowledge flow. Dalmarco, in turn, conducted studies that analyze knowledge flow in joint projects between universities and enterprises [30,44,132] which foster a favorable environment for this type of interaction. Table 5 mentions the factors and authors contributing to technological transfer performance from the university–industry relationship.

**Table 5.** Dimensions and factors of the university–industry relationship.

| Dimensions | Factors | Authors |
|---|---|---|
| Liaison mechanisms and units between the university and the environment | Transfer channels | [30,44,128,133–135] |
| | Transfer strategies | [12,20,26,56,82,136] |
| | Knowledge flow from the university and industry | [6,26,30,44,50,96,132,137–140] |
| | Common interests | [117,141] |
| | Communication processes | [117,141] |
| | Strategic alliances | [56,64,85] |
| | Liaison units (internal, external, or mixed) | [142] |
| | Collaborative processes | [19,143] |
| | Insertion of HEIs into the productive environment | [97,126,144–146] |
| Characteristics of the players intervening in technological transfer | Researcher profile | [80,82,136,147–149] |
| | Researcher's position on the transfer | [12,26,99,136,143,147,149] |
| | University profile | [16,97,105,150] |
| | Profile of the liaison unit (research results transfer office) | [7,26,77,91,99] |
| | Profile of enterprises receiving research results from academia | [78,80] |

Concerning the results obtained in the VOSviewer software application from a Research Information System (RIS) file generated from ZOTERO, analyses focused on the correlation of keywords and authors were carried out using the visualization of similarities

(VOS) mapping technique and the clustering technique where a cluster is a set of closely related nodes and, according to the type of link being analyzed, each node is assigned precisely to a cluster. Additionally, analyses were performed to describe the publication ratio per year and the sources/publications where more articles related to the study object were found. These results are summarized in Figures 3–6.

Figure 3 shows the VOSviewer network map of the keywords with the highest occurrence. The circle size related to the word is proportional to the frequency of occurrence (number of documents in which a keyword appears). In the network, the degree of similarity they may have with other keywords is also illustrated using lines. The words with the highest frequency of appearance in the papers stand out: technology transfer, knowledge transfer, university technology transfer, knowledge flow, innovation, collaborative research, business university, academic entrepreneurship, dynamic capabilities, and business university links, among others.

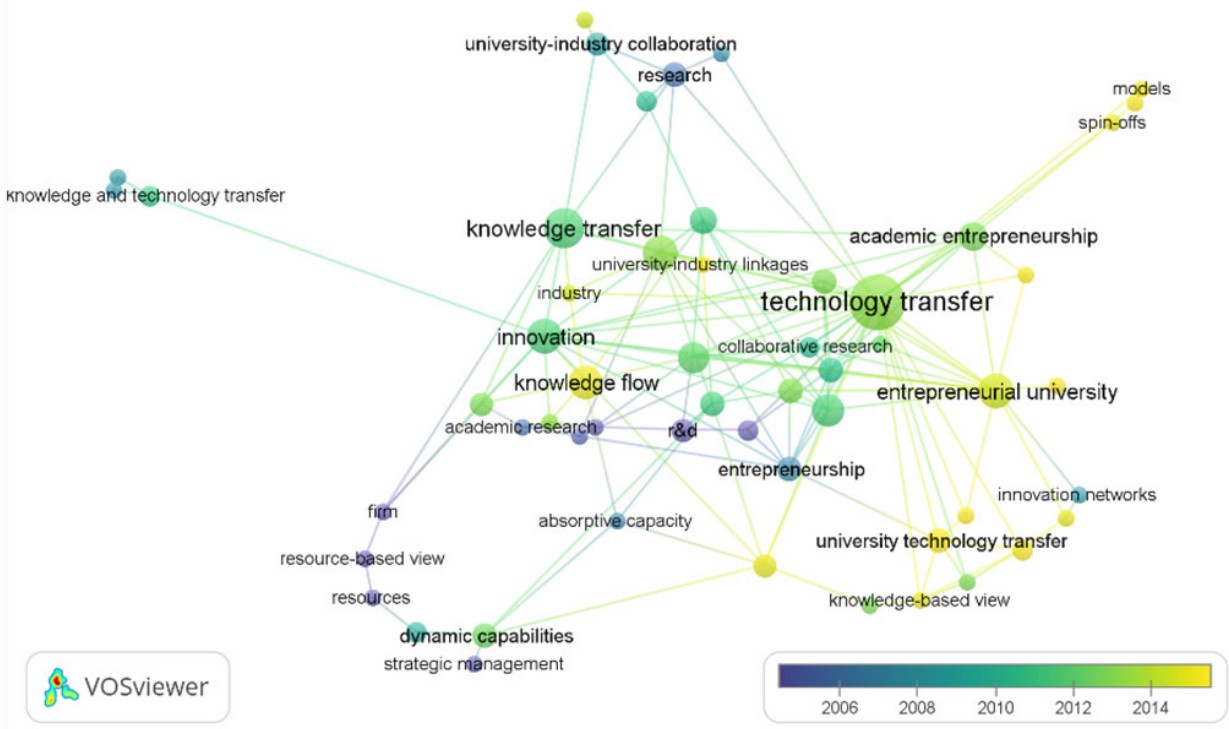

**Figure 3.** VOSviewer network map of keyword occurrence. Source: elaborated by the authors based on results from VOSviewer Version 1.6.11.

Figure 4 shows a map of the co-authorship network where the authors who appear most frequently in co-authorship are listed. The authors with the most significant occurrence in the articles are highlighted with the size of the relevant circle, and stand out as the authors who have contributed the most significant number of articles. Papers in the field of technology transfer with a knowledge-flow approach have been written by Dalmarco, Etzkowitz, Teece, D'este, and Bozeman, among others.

Likewise, Figures 3 and 4 display a visualization scale that relates the parameters analyzed with the year of publication using a color scale and differentiating the cluster generated according to the relationship or affinity presented.

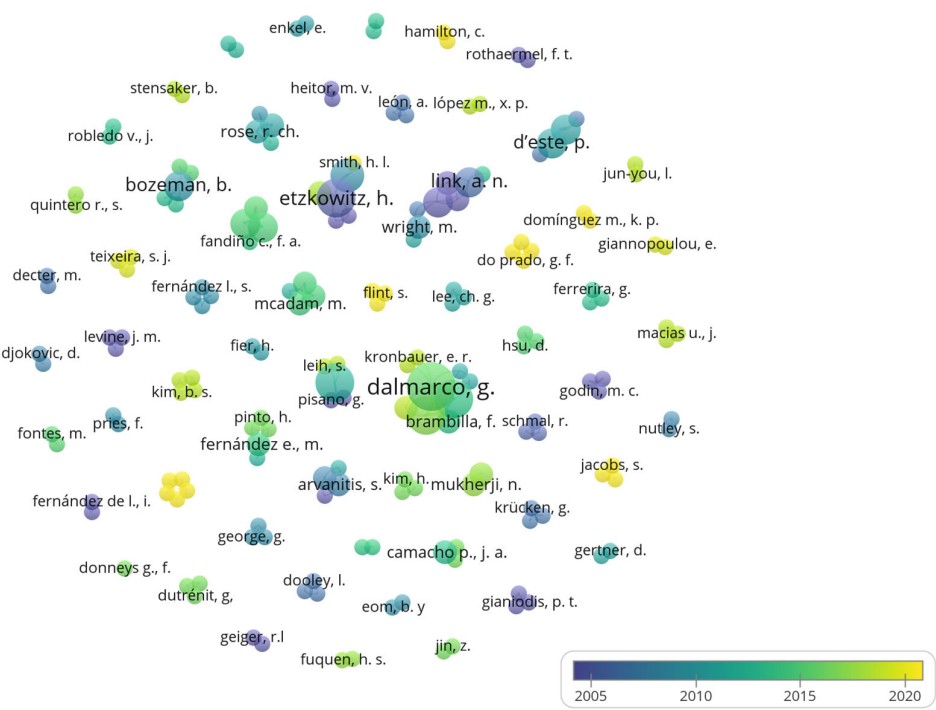

**Figure 4.** Co-authoring VOSviewer network map. Source: elaborated by the authors based on results from VOSviewer version 1.6.11.

Figure 5 shows the production per year of the articles on technological transfer in universities with a knowledge-based vision which were finally selected, highlighting an increase in 2013 and 2018.

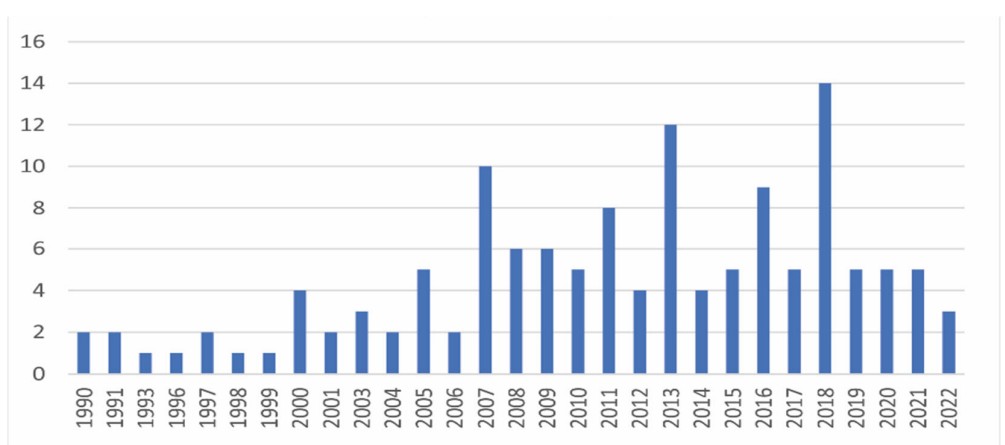

**Figure 5.** Articles per year. Source: elaborated by the authors.

Figure 6 relates the journals or publications where authors publish articles with themes related to technological transfer.

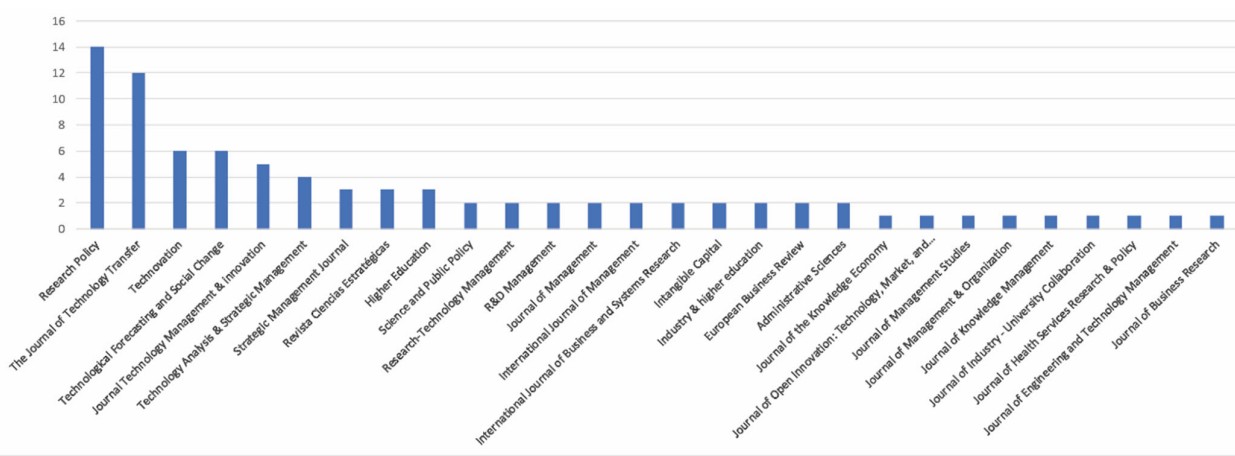

**Figure 6.** Journals/publications. Source: elaborated by the authors.

Finally, Figure 7 illustrates the number of articles analyzed per conceptual axis, indicating that some of the 135 articles selected identified factors related to one or more conceptual axes or dimensions.

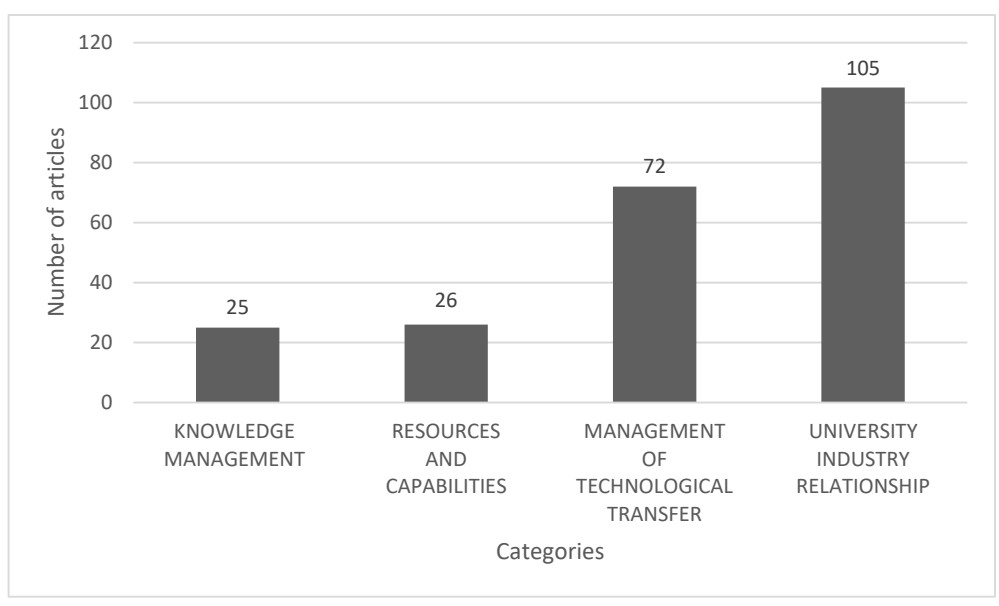

**Figure 7.** Relationship between the number of articles by conceptual axis. Source: elaborated by the authors.

## 5. Discussion

In order to identify the factors in the academic literature that contribute to technology transfer in universities from a knowledge-flow perspective, the elements discussed by Dalmarco et al were considered [30]. Under this perspective, we can conceive of the knowledge flow generated in the relationship of the university with the other actors as having two aspects: whoever generates or initiates the transfer (direction) and that which will be transferred (knowledge content). After reading and analyzing the 135 documents and using the axial coding criteria given by Hernández et al. [52], four axes upon which the discussion will be based were identified: knowledge management, resources and capabilities, technological transfer management, and the university–industry relationship. We identified how the literature on technological transfer in universities had been enriched through the contribution of theory based on resources and capabilities [71,75,76,151] and a knowledge-based vision [34,44,152], evidencing the role of knowledge management and

knowledge flow as a source of dynamic capabilities [62,70] and their contributions to the university–industry relationship [15,16,106].

### 5.1. Knowledge Management

Since the emergence of the knowledge-based economy in the 1990s, a significant change has been introduced in conceiving technology transfer [145,146]. The traditional model centered on a well-defined technology that moves from a specific economic unit (company department, laboratory, enterprise, or country) to another established economic unit. The knowledge-based approach suggests an entirely new technology transfer model, where interest has mainly shifted to analyzing the interactions among the various actors involved in the technological innovation process and, therefore, in the transfer [34]. It has highlighted how the knowledge-based economy not only promotes the development of technologies, methodologies, and strategies in knowledge-based enterprises but also how its measurement, creation, and diffusion, generates that knowledge and becomes one of the main priorities of organizations and an essential element for economic and social development [153]. This shows the importance of bearing in mind the factors that intervene in the creation [28,55,56,58], storage, transfer [21,29,54,57–59,63,65], application and use of knowledge [36,57,60–62,67–70] from the universities.

### 5.2. Resources and Capabilities

Concerning resources and capabilities, it is pointed out how authors have managed to identify the interest the players have shown not only in identifying, evaluating, and quantifying resources [7,21,61,74–78,81,82,84,85,154] and capabilities [21,34,62,69–72,79,153] related to the organization's knowledge, but also in how these factors can speed up internal R&D processes and encourage other players to see them as strategic allies in transfer processes, leading to collaborative research, development, and innovation processes. In the same way, the interest that the subject of intellectual property has aroused is highlighted [12,26,65,87,88]. This is fostered by the concern expressed by different institutions and/or players involved in the transfer process, its regulation and compliance that guarantee and promote the best environment to ensure that technologies generated from research processes are finally commercialized. It was shown that knowledge flow is considered a fundamental element in potentiating dynamic capacities.

### 5.3. Technological Transfer Management

From technological transfer management and all this implies, the role Technology Transfer Offices (TTO) and Research Results Transfer Offices are fulfilling today is highlighted [11,23,46–49]; their work to support the research process to promote business innovation is the foundation of the technology transfer process in universities, strengthens the third mission of university "extension", and promotes and articulates knowledge and/or technology transfer to the production or social organizations that require it [11,23,47–49]. It is crucial in this process to identify the actors, functionalities, and responsibilities (knowledge flow direction) and, of course, to be clear about what is to be transferred (knowledge flow content), thereby achieving more effective management.

For this reason, universities seeking to improve the performance of technology transfer must, among other actions: promote interaction and exchange of knowledge with actors in the innovation system [21,23,77,102,104], define their capacities for research and extension [66,77,103], and articulate the dissemination and transfer of knowledge within society and the productive sector [3,10,155], strengthen the offices, units or divisions in charge of technology transfer within the institution (OTRIS-OTT) and adopt innovation processes [8,106,155,156] promoting the creation of new technology-based companies [49,115,142].

*5.4. University–Industry Relationship*

In relation to the university–industry relationship, we have noted the different contributions that have been made by the Triple Helix model [59,97,106,130] and its evolution in search of commercializing the results from research taking place in universities and fostering the entrepreneurial environment in future universities [16,97,105,108,157]. Within this context, open innovation emerges as an alternative that opens new possibilities for the commercial exploitation of knowledge in universities [158,159], accelerating the innovation process and propitiating two-way technology or knowledge transfer, both internal and external. Furthermore, players such as society and the environment that play an essential role in technology transfer processes appear from the conception of the quadruple and quintuple helix [64,122,123].

Finally, some characteristics are mentioned which are considered important in technology transfer from a knowledge-flow approach:

- The transfer is not a process defined explicitly by signing a contract such as a license or a joint R&D + i development agreement.
- The transfer process must specify the functions of the players (direction of knowledge) and the purpose of the transfer (the content of knowledge).
- Intellectual property specifications (policies and rights and duties acquired) should be proposed.
- Technology transfer must be a collaborative process, where the technology donor and recipient understand that a success transfer occurs when the technology is used by the recipients in their environment, fulfilling the required or manifested need from the beginning of the process.
- Technology transfer is a process that does not end until the recipent of technology adopts it according to agreed performance indicators.
- Within the university, the transfer must be considered a strategy which is part of the third mission that permits solving problems of the environment and generating economic benefits for the institution and the technology recipient.
- Universities must optimize their resources and capabilities to benefit from research processes and establish guidelines that permit the benefactors of technological institutions to increase their absorption capacity.
- Technological transfer processes must have trained human resources that are open to change and encouraged to motivate and coordinate the process to benefit the players involved.
- Communication, understanding, and trust among the players are fundamental in technological transfer processes.
- There must be mechanisms, policies, and/or principles that foster trust and collaborative work so that the performance of technological transfer ends in good conditions for the parties involved.

## 6. Conclusions

According to work presented, the results obtained, and their discussion, the following main conclusions may be arrived at:

It is possible to classify four conceptual axes permitted identifying and systematizing the different factors that—from a knowledge-flow approach—can contribute to technological transfer processes in universities.

The factors identified from a knowledge-based vision permit the knowledge flow that takes place in the university–industry relationship to be defined and strengthened. This enables the commitments by the players who intervene in the TT process (direction) to be specified and established, providing clarity and propitiating tools which allow what is going to be transferred and under what conditions (content) the TT will be carried out to be established. All the preceding supposes better performance of the technological transfer is reflected in the efficiency and effectiveness of the process intervening in this process.

Knowledge-based vision has permitted the creation of new organizational models that facilitate and dynamize the participation and insertion of Higher Education Institutions (HEIs) in the productive setting, highlighting the role of the university as a key player in innovation processes, as a source of new knowledge codified through research, and a provider of high-level human capital.

Based on the analysis, interpretation, and implementation of the identified factors, a large number of HEIs have strengthened and diversified their knowledge and technology transfer function, complementing the teaching, research, and extension process, promoting greater participation and significant advances in innovation issues, managing to boost business capacities and the creation of new technology-based companies.

Consequently, our findings synthesize and map in an integral way based on the conceptual framework of the study of TT, identifying a group of factors that, from the flow of knowledge, do not only contribute to analyzing the relations between the industry and the university but also broaden the discussion and complement the studies by Dalmarco [30,44,132] concerning the flows of knowledge that occur in the Transfer of technology.

Interest was identified in the scientific community in contining to contribute to issues related to technology transfer in order to strengthen the articulation between research and its results and the coproduction of knowledge with the actors of the fourth helix, consolidating the third mission not only from the application of knowledge but also from an adequate flow of knowledge that allows the performance of technology transfer from universities to be strengthened. Universities must continue to be strategic allies based on knowledge, research, and development and in the company of the government, the community, and the productive sector.

Future work should delve into absorption capacity issues concerning companies that have ties or see universities as strategic allies, as well as into innovation themes, particularly open innovation, to identify elements that—from their conception—can contribute to technology transfer in universities from the knowledge-flow approach.

Although this systematic review of the literature allowed us to identify that companies increasingly focus on cooperation with universities, a methodology must be designed which can become a roadmap for the identified factors that improve the performance of technology and knowledge transfer from universities to enterprises.

**Author Contributions:** Conceptualization, J.B.P.B., J.W.Z.S., C.O.-L. and M.R.-C.; methodology, J.B.P.B. and J.W.Z.S.; validation, J.B.P.B., J.W.Z.S., C.O.-L. and M.R.-C.; formal analysis, J.B.P.B.; investigation, J.B.P.B.; resources, J.B.P.B.; data curation, J.B.P.B. and J.W.Z.S.; writing—original draft preparation, J.B.P.B., J.W.Z.S., C.O.-L. and M.R.-C.; writing—review and editing, J.B.P.B., J.W.Z.S., C.O.-L. and M.R.-C.; visualization, J.B.P.B.; supervision, J.W.Z.S. All authors have read and agreed to the published version of the manuscript.

**Funding:** This research received no external funding.

**Institutional Review Board Statement:** Not applicable.

**Informed Consent Statement:** Not applicable.

**Data Availability Statement:** Not applicable.

**Conflicts of Interest:** The authors declare no conflict of interest.

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
