# Peer review of "University Technology Transfer from a Knowledge-Flow Approach—Systematic Literature Review"

_sustainability, doi:10.3390/su15086550_

Round 1

Reviewer 1 Report

1. Kindly refer to the first sentence in Line 47. Would be better if rewritten in a more understandable way.

2. Refer to the statement in Line 66 “Other authors focus on the transfer channels and mechanisms [15,19], 66 as well as some that describe the barriers that emerge in said relation and which affect the 67 technology transfer [17,19–26].”. Kindly elaborate.

3. Figure 2 and Figure 3 have to be presented more clearly. Lot of confusion is there in both the figures.

Reviewer 2 Report

The authors spent a lot of time looking up a lot of literature, which is very commendable! Such a wealth of literature has allowed me to gain a lot of new knowledge.

Such rich literature provides a good foundation for the smooth development of this research, but it also increases the difficulty of the research, that is, how to effectively summarize and analyze so much literature. The ideas and methods are more suitable.

However, what is the purpose of this article? It doesn’t seem to be well articulated. It determines which kinds of literature the authors searched.

The conclusion needs to be rewritten. In particular, it may not be necessary to recite sources that have already been explored. After analyzing so much literature, the author is fully able to condense some core ideas. Future research plans should also echo the purpose of the study or the problems that the study did not address. At the same time, research whether there are vulnerabilities. Whether there are limitations or not, this also needs to be well explained by the authors.

The chapters are numbered incorrectly. These omissions can easily make the reader feel that the author’s attitude when writing the article is not very rigorous. It is necessary to carefully proofread the manuscript.

Reviewer 3 Report

The introduction needs connectivity to innovation adoption, stakeholder/actor openness to innovations, and knowledge transfer. 

I recommend included the following scholarship to assist in filling this gap in the manuscript:

Innovation adoption - https://doi.org/10.1016/j.agsy.2020.102908

technology salience for transfer - https://doi.org/10.3390/s22186833

Knowledge transfer - https://doi.org/10.1016/j.agsy.2014.02.005

Authors provided a robust framework that was indicative of alignment with the introduction/need for the study. This reviewer was intrigued authors did not include Rogers' (2003) diffusion of innovations. Please add in the theoretical framework the reasons or extent DoI was not included or why the authors felt DoI was not appropriate. These justifications would be important to the literature and international scholars as a whole. 

Please add to the merits, strengths, and weaknesses of a systematic review and why a systematic review was chosen versus a scoping review. See Wright et al. (2007) here https://doi.org/10.1097/blo.0b013e31802c9098 or other Sustainability authors implementing a systematic review https://doi.org/10.3390/su15054524, https://doi.org/10.3390/su15053949, or https://doi.org/10.3390/su131810295. The methodology needs more details regarding the systematic review. 

The findings section is the most robust attribute of the scholarship based on the objectives provided. The figures and illustrations provided are an asset to readers and future scholars. 

Beginning at Section 4 going through Section 5, more support, connectivity, and recommendation for practitioners are necessary. See the feedback in the Introduction. When that section is enhanced, authors will be able to increase the rigor, scholarship, and practical implications for the actors this scholarship improves to a higher degree. 

The manuscript has a lot of merit but the Introduction, Methodology, Discussion, and Conclusion warrant increased quality and expansion. 

Round 2

Reviewer 2 Report

The changes made by the authors basically solve the problems that existed in the past, which deserve recognition!

In order to better improve the quality of the article, I recommended the authors double-check all the literature and relevant cases to ensure that they are correct and credible.

At the same time, it is necessary to proofread the article again.

Author Response

Again we appreciate your comments. Seeking to improve the quality of the paper, we accept the recommendation and review the document and adjust or add a couple of references.

Reviewer 3 Report

The authors greatly improved the scholarship. 

Though the connectivity to the asset of extension or outreach systems at institutions can bolster the dissemination of knowledge transfer and thus, impact to actors or players is still lacking in the Conclusion - in my opinion. 

Not conveying those potential merits in knowledge transfer of insitution's capacity will limit the future citations of this work from the thousands globally that investigate knowledge transfer within community leaders and actors. 

The authors made an attempt to address my previous feedback here "Based on the analysis, interpretation and implementation of the identified factors, a large number of HEIs have strengthened and diversified their knowledge and technology transfer function, complementing the teaching, research and extension process, promoting greater participation and significant advances in innovation issues, managing to boost business capacities and the creation of new technology-based companies." but from the lens of localized practitioners, there is not information supporting the why, how, and when. Additionally, the above paragraph is not supported by any of the citations i recommended in my previous review. 

This scholarship answers the "So what?" question for science but does not answer the "So what?" question for practitioners in terms of how does the scholarship improve practice. This is still lacking in my opinion and a barrier for the holistic scholarship of the manuscript. 

Author Response

Thank you again for your comments, very pertinent, by the way. Seeking to improve the quality of the article, we accepted the recommendation and added information with their respective references in the discussion, particularly in section 5.3, seeking to support and complement what was mentioned in the conclusions.

5.3 Technological transfer management

From technological transfer management and all this implies, the role Technology Transfer Offices (TTO) and Research Results Transfer Offices are fulfilling today is highlighted [11,23,48–51]; their work to support the research process to promote business innovation is the foundation of the technological transfer process in universities, strengthens the third mission of the university “extension”, promotes and articulates knowledge and/or technology transfer to production or social organizations that require it [11,24,50,51,116]. It is important in this process to clearly identify the actors, functionalities and responsibilities (knowledge flow direction) and of course to be clear about what is to be transferred (knowledge flow content), achieving more effective management. It is important in this process to clearly identify the actors, functionalities and responsibilities (knowledge flow direction) and of course to be clear about what is to be transferred (knowledge flow content), achieving more effective management.

For this reason, universities seeking to improve the performance of technology transfer must, among others: promote interaction and exchange of knowledge with actors in the innovation system [21,24,78,103,105], define their capacities research and extension  [67,78,104] and articulate the dissemination and transfer of knowledge with society and the productive sector [3,10,158], strengthening the offices, units or divisions in charge of technology transfer within the institution (OTRIS - OTT) and innovation adoption processes [8,107,158,159] promoting the creation of new technology-based companies [116,117,145].

Round 3

Reviewer 3 Report

The authors have greatly improved the manuscript since the first submission. Revisions provided in this third review round now warrant the publication of this scholarship in the journal. 

I believe you have sufficiently revised the manuscript for publication.